# Role of Epigenetics for the Efficacy of Cisplatin

**DOI:** 10.3390/ijms25021130

**Published:** 2024-01-17

**Authors:** Tatjana Lumpp, Sandra Stößer, Franziska Fischer, Andrea Hartwig, Beate Köberle

**Affiliations:** Department Food Chemistry and Toxicology, Institute of Applied Biosciences, Karlsruhe Institute of Technology, Adenauerring 20a, 76131 Karlsruhe, Germany; tatjana.lumpp@kit.edu (T.L.); sandra.stoesser@kit.edu (S.S.); franziska.fischer@kit.edu (F.F.); andrea.hartwig@kit.edu (A.H.)

**Keywords:** epigenetics, promoter methylation, histone modification, miRNA, lncRNA, cisplatin resistance

## Abstract

The clinical utility of the chemotherapeutic agent cisplatin is restricted by cancer drug resistance, which is either intrinsic to the tumor or acquired during therapy. Epigenetics is increasingly recognized as a factor contributing to cisplatin resistance and hence influences drug efficacy and clinical outcomes. In particular, epigenetics regulates gene expression without changing the DNA sequence. Common types of epigenetic modifications linked to chemoresistance are DNA methylation, histone modification, and non-coding RNAs. This review provides an overview of the current findings of various epigenetic modifications related to cisplatin efficacy in cell lines in vitro and in clinical tumor samples. Furthermore, it discusses whether epigenetic alterations might be used as predictors of the platinum agent response in order to prevent avoidable side effects in patients with resistant malignancies. In addition, epigenetic targeting therapies are described as a possible strategy to render cancer cells more susceptible to platinum drugs.

## 1. Introduction

Cisplatin and its analogs carboplatin and oxaliplatin (Figure 1) are platinum-containing drugs that are widely used in cancer therapy. Cisplatin (cis-diamminedichloro-platinum(II)) is a major component in the treatment of a wide variety of malignancies, as confirmed by various former and recent clinical trials [1,2,3,4]. Cisplatin is most effective, particularly for patients with testicular germ cell tumors or ovarian cancers, but it also displays clinical activity against bladder, prostate, head and neck, cervical, breast, and lung cancers [5,6,7,8,9]. Carboplatin shows a comparable mode of action and is used as first-line treatment for patients with advanced ovarian cancer, but this platinum-based drug is also applied for the treatment of a number of other types of cancer, such as advanced small-cell and non-small-cell lung cancer [10]. In contrast, oxaliplatin has a distinct pattern of activity. For the treatment of metastatic colorectal cancer, particularly for cases insensitive to cisplatin and carboplatin, oxaliplatin is used in combination with 5-fluorouracil [11].

The mechanisms underlying the antitumor activity of cisplatin have been studied extensively in various cultured tumor cell lines and experimental animal models, yielding compelling evidence that the major toxic mode of action is mediated by its interaction with DNA, leading to lesions that will disrupt the structure of the DNA molecule [12,13]. Cisplatin is actively taken up by cells via several membrane transporters but also enters the cells by passive diffusion through the plasma membrane [14,15]. Once inside the cell, cisplatin is activated by the replacement of the two labile chloride-leaving groups with water [10]. Activated cisplatin will subsequently react with nucleophilic residues of DNA, particularly with N7 of purines, leading to different types of DNA lesions. The main DNA lesions induced by cisplatin are intrastrand DNA crosslinks between two purine bases on the same DNA strand, while interstrand crosslinks (ICLs) between two guanines on the opposite DNA strands are formed to a lesser extent [8,12,16,17,18]. Cisplatin lesions impair normal DNA functions, leading to the initiation of multiple signal transduction pathways, which either induce cell cycle arrest, allowing cells to repair the damage or, if the damage cannot be repaired, trigger apoptosis and hence cell death [5,6,8,11,12]. In addition to damaging DNA, cisplatin is also known to induce reactive oxygen species, which in turn will result in the activation of apoptotic pathways and hence also contribute to cisplatin cytotoxicity [19].

Despite its activity against various tumors, resistance towards cisplatin remains a major clinical problem that limits the therapeutic success with poor prognosis. Resistance may be intrinsic to a tumor, as itis the case for prostate cancer and colorectal carcinoma, or it might be acquired during courses of chemotherapeutic treatment, as observed, for example, in ovarian cancer. The mechanisms underlying cisplatin resistance appear to be multifactorial. Intrinsic resistance is often associated with the inactivation of the DNA mismatch repair process (MMR), which is observed in inherently cisplatin-resistant colorectal cancer. Acquired cisplatin resistance has been extensively studied in experimental models of cisplatin-resistant cell lines. Resistance factors include reduced intracellular cisplatin accumulation due to changes in drug transport, increased drug detoxification as a result of elevated levels of intracellular scavengers such as glutathione and/or metallothioneins, altered DNA repair mechanisms, namely, nucleotide excision repair, ICL repair and DNA mismatch repair (MMR), and alterations in the apoptotic signal transduction pathways, which enable cancer cells to evade cisplatin-induced cell death [12,20,21,22]. Furthermore, mechanisms involving epigenetic regulation are increasingly recognized to affect cellular sensitivity to cisplatin treatment (Figure 2) [23,24].

Epigenetics refers to alterations in gene expression that are stable, reversible, and heritable and occur without any change in the DNA sequence [25,26]. The main epigenetic mechanisms associated with chemosensitivity and chemoresistance are (i) DNA methylation, (ii) histone modification and chromatin remodeling, and (iii) gene expression regulation through RNA interference mediated by non-coding RNA such as microRNA (miRNA) or long non-coding RNA (lncRNA) (Figure 3). Modifications to genes and their respective chromatin structures, specifically through DNA methylation and histone modification, may affect the accessibility of the transcriptional machinery to DNA, consequently influencing transcription without any changes in the nucleotide sequence of the respective gene, while microRNAs regulate gene expression on the post-transcriptional level. In this review, we address the epigenetic mechanisms associated with cisplatin efficacy and discuss possible strategies to sensitize cisplatin-resistant tumors using epigenetic targeting. In addition, epigenetic changes might be used as predictors of the platinum agent response to avoid unnecessary side effects in patients with tumors resistant to cisplatin treatment.

## 2. Epigenetic Modifications

This section provides a concise overview of individual epigenetic signaling pathways and their currently discussed involvement in cisplatin resistance. Each epigenetic regulation mechanism is assigned its own subsection. In the beginning, a brief presentation of an epigenetic pathway is given, followed by a detailed description of its underlying mechanism and its specific role in the development of cisplatin-induced tumor resistance.

### 2.1. DNA Methylation

One of the key epigenetic pathways is the DNA methylation process. Here, the addition or removal of a methyl group on DNA serves to alter the transcriptional regulation machinery in a gene-specific manner.

#### 2.1.1. Mechanism of DNA Methylation

Methylation of DNA occurs mainly at a cytosine followed by a guanine in so-called CpG dinucleotide sites [27]. DNA methylation is catalyzed by DNA methyltransferases (DNMTs) which transfer methyl groups to the carbon 5-position of cytosine, using *S*-adenosyl methionine (SAM) as a donor for the methyl group [28]. The DNMT family of enzymes comprises a number of isoforms, including DNMT1, DNMT3a, and DNMT3b, which are the three major DNMTs. DNMT1 is the most prevalent DNMT enzyme and is considered a maintenance methyltransferase as it preferentially binds to hemi-methylated DNA and ensures the methylation pattern after DNA replication by transferring the methyl group to the daughter strand. DNMT3a and DNMT3b act mainly as de novo DNA methyltransferases for initial DNA methylation [27,28,29].

CpG sites are often clustered in CpG islands. These stretches of CpG-rich regions are defined as sequences of more than 200 bp with G and C of ≥50%, of which at least 60% are positioned in the CpG content [30]. CpG islands are typically located within the vicinity of gene promoters or other regulatory sequences required for gene transcription [31]. The DNA methylation status appears to be essential for proper gene regulation, as hypermethylation of CpG islands within a gene promoter results in condensed DNA with reduced accessibility for the transcription machinery and, hence, gene silencing. For example, this is often observed for genes implicated in carcinogenesis such as tumor suppressor genes. DNA hypomethylation of the promoter, on the other hand, appears to be associated with the activation of gene expression (Figure 4) [32,33,34,35,36]. For instance, it has been observed that DNA hypomethylation of promoter regions of oncogenes resulted in an activation of the respective gene, which has been associated with tumor development. DNA methylation may directly interfere with the binding of transcription factors to the promoter region, or it may suppress transcription via the binding of methyl-CpG-binding proteins, such as MeCP2, which contains a transcriptional repression domain capable of silencing gene expression [34,35,37].

The cellular methylation status is also regulated by expression and activity of nicotinamide N-methyltransferase (NNMT) [38]. NNMT is an enzyme that catalyzes the methylation of nicotinamide to N-methyl nicotinamide using SAM as a methyl group donor. It has been observed that NNMT expression and activity are frequently elevated in tumor tissue, causing a decrease in the cellular SAM content, which in turn will lead to hypomethylation of genes associated with tumor progression and metastasis. As NNMT supports tumorigenesis, it therefore may be considered a potential therapeutic anticancer target. Indeed, various molecules that target NNMT have been identified, which might be applied to overcome chemoresistance [39,40,41].

#### 2.1.2. DNA Methylation and Cisplatin Resistance

Aberrant changes in DNA methylation have been implicated in various diseases. Global genome DNA hypomethylation together with tumor suppressor gene hypermethylation is frequently observed in cancers, indicating a role of epigenetics in tumor development [33,35]. For example, tumor suppressor genes *p16* and *RASSF1A* are frequently silenced in various types of cancer, and it is thought that the subsequent disturbance of pro-apoptotic pathways contributes to increased proliferation and drug resistance [35,42,43,44]. Along with its involvement in tumor development, there is considerable evidence that aberrant DNA methylation has clinical implications for tumor therapy [45,46]. Various findings suggest that an increase in DNA methylation may be linked to cisplatin resistance [45,46,47,48]. Integrated analysis of DNA methylation and mRNA expression profiling revealed a panel of candidate genes involved in cell cycle progression and apoptosis showing higher methylation levels in cisplatin-resistant primary tumor samples of non-small-cell lung cancer (NSCLC) compared with cisplatin-sensitive NSCLC [49]. An in vitro model of ovarian cancer cell lines showed that acquired drug resistance was accompanied by hypermethylation of a large number of CpG islands [50]. In addition, cisplatin-resistant tumor cell lines can exhibit a microsatellite instability phenotype, which is a hallmark of a defective cellular DNA MMR system [51]. This repair pathway deals with single base mispairs or looped intermediates that arise during replication or as a result of DNA damage. MMR is initiated by specific MMR complexes, with MMR complex MSH2-MSH6 preferentially binding to single base mispairs and loops of one to two bases and MSH2-MSH3 recognizing loops larger than two bases [52,53]. The complex MLH1-PMS2 is then recruited, followed by excision of the damage and re-synthesis of the excised strand [53,54]. Cells deficient in MMR are considerably more tolerant to cisplatin treatment, indicating an association between a loss of the MMR system and platinum resistance [55,56,57,58].

An early investigation revealed a lack of expression of the MMR gene *MLH1* in tissues of colon cancer due to *MLH1* promoter methylation, whereas adjacent normal tissue expressed the MLH1 protein, suggesting a relationship between *MLH1* epigenetics and colorectal cancer [59]. Based on these observations, several studies have used in vitro models of various cancers to investigate a possible association between the loss of MMR proteins due to DNA hypermethylation and resistance to chemotherapeutic drugs, e.g., cisplatin. In cisplatin-resistant sublines independently derived from the ovarian cancer cell line A2780, increased methylation of the *MLH1* promoter, resulting in loss of MLH1 expression, was observed, and the authors concluded that hypermethylation-dependent inactivation of MMR emerged in a cisplatin-resistant phenotype [60]. This was supported by investigations using the epigenetic modifier 5-azacytidine (5-aza), a de-methylating agent, which re-activated *MHL1* expression and restored cisplatin sensitivity in the A2780 sublines. Hypermethylation of the *MLH1* promoter and loss of protein expression was also shown in a significant fraction of clinical samples of ovarian tumors, wherein the suggestion was made that hypermethylation of *MLH1* could be a contributing factor to acquired platinum resistance, which is frequently observed in ovarian carcinoma. DNA isolated from plasma of patients suffering from ovarian cancer has been used to monitor CpG island methylation changes, and it was observed that methylation of the *MLH1* promoter increased at relapse and predicted a poor clinical outcome following different chemotherapy regimens, including the platinum drug carboplatin, which induces the same DNA lesions as cisplatin [61]. The methylation status of the *MLH1* gene might therefore be suitable as one predictor of response to platinum-based chemotherapy for this type of cancer. This is supported by a DNA methylation analysis, which revealed a number of genes, among them *MLH1*, that were hypermethylated in a human ovarian cancer cell line model of cisplatin resistance, suggesting the utilization of these genes as clinically relevant biomarkers [62]. However, no relationship between *MLH1* methylation-mediated loss of protein and resistance to systemic cisplatin treatment could be demonstrated in testicular cancer tissue [63], suggesting that the relevance of *MLH1* methylation for cisplatin resistance might be tumor-type specific.

With the exception of MMR genes, CpG island methylation of the genes involved in DNA repair is generally associated with an improved response to cisplatin treatment, as shown by Teodoridis and co-workers in samples of late-stage ovarian tumors [64]. Several studies have sought to establish whether methylation of the *BRCA1* promoter modulates cisplatin sensitivity using in vitro and in vivo models. BRCA1 is a tumor suppressor protein, which is crucial for the homologous recombination-dependent repair of DNA double-strand breaks and ICLs [65]. Due to their reduced ability to repair ICLs, BRCA1-deficient cell lines are susceptible to ICL-inducing agents such as cisplatin [66,67,68]. An increased sensitivity toward cisplatin has also been observed in animal models with BRCA1-deficient mammary tumors [69,70]. Reduced or complete loss of BRCA1 protein expression is observed as a result of *BRCA1* germline mutations and also of *BRCA1* promoter methylation [71]. Epigenetic inactivation of *BRCA1* has mainly been observed in breast and ovarian carcinomas. In breast cancer cell lines with *BRCA1* promoter methylation, low expression levels of BRCA1 mRNA and protein were observed, and these cell lines showed cisplatin sensitivity comparable to that of *BRCA1* mutated cells [72]. The cellular response to cisplatin has also been assessed in an in vivo model of xenografts of *BRCA1*-methylated and unmethylated breast cancer cells, which showed that xenografts of *BRCA1*-methylated cells were significantly smaller following cisplatin treatment than xenografts of unmethylated cells [72].

Translation of these preclinical findings to clinical samples showed that *BRCA1* promoter methylation was associated with lower BRCA1 mRNA levels in breast cancer tumor specimens [73]. Therefore, the clinical relevance of *BRCA1* promoter methylation for the response to cisplatin chemotherapy in vivo has been investigated both in tumor xenografts and in clinical studies. Patient-derived xenograft models of triple-negative breast cancer (TNBC) were used to investigate a possible correlation between *BRCA1* methylation status and cisplatin sensitivity [70]. TNBCs are negative for estrogen and progesterone receptors and show a lack of overexpression and/or gene amplification of the human epidermal growth factor receptor 2 (HER2). TNBC accounts for 15–20% of breast cancers, with a substantial percentage displaying a deficiency in BRCA1 [74]. Treatment of TNBC xenograft models with chemotherapeutic drugs, including cisplatin, showed prolonged overall survival in models lacking BRCA1 mRNA and protein due to *BRCA1* promoter hypermethylation [70]. The authors also observed an association between *BRCA1* promoter methylation and a favorable response to cisplatin chemotherapy in TNBC patients, indicating the clinical relevance of the methylation status of *BRCA1* to cisplatin therapy. This is supported by findings of Silver and co-workers, who observed a better response to cisplatin in patients with TNBC when the *BRCA1* promoter was hypermethylated, and hence, mRNA levels were lower in the tumor tissue compared with tumors without *BRCA1* promoter methylation. Even though no complete response could be observed in the patient groups, the authors suggested that a subset of TNBC may be sensitive to cisplatin therapy based on the methylation status of *BRCA1* [75]. In a clinical cohort of ovarian cancer, epigenetic inactivation of *BRCA1* was linked to a better response to cisplatin treatment and increased overall survival, suggesting that *BRCA1* methylation affects the efficacy of cisplatin therapy and hence, the clinical outcome in this type of tumor [72]. Based on these observations, one might suggest that *BRCA1* promoter methylation might be used as a prognostic biomarker to recognize breast or ovarian cancer patients who might benefit from cisplatin treatment.

The BRCA1 protein is also involved in the Fanconi anemia (FA) pathway. The FA/BRCA pathway deals with DNA ICLs and hence, controls the cellular efficacy of DNA ICL-inducing agents, such as cisplatin [76]. As previously mentioned, cisplatin is used for the treatment of ovarian cancer, which is initially sensitive to the drug but might acquire resistance during treatment over time. Observations in tissues of primary ovarian cancer revealed that the FA gene *FANCF* was methylated in 21% of the tissues examined, and it was hypothesized that the disruption of the FA/BRCA pathway might contribute to the initial sensitivity toward cisplatin. Ovarian cancer cell lines were therefore used to investigate whether *FANCF* methylation is associated with cisplatin sensitivity, and these cisplatin-sensitive cells exhibited a densely methylated promoter region of the *FANCF* gene, which was accompanied by a lack of *FANCF* mRNA expression. Retroviral transduction with *FANCF* cDNA resulted in protein expression and the partial restoration of cisplatin resistance [77]. Other studies found *FANCF* promoter methylation in oral and lung cancer, suggesting disruption of the FA/BRCA pathway also in these types of cancer [78]. Based on these observations it can be discussed that patients with epigenetic inactivation of the FA/BRCA pathway might benefit from treatment with ICL-inducing chemotherapeutics, including cisplatin, and *FANCF* promoter methylation might serve as a predictive marker.

Promoter methylation of the *PAX5* gene has also been identified in connection with cisplatin resistance, both in vitro models and in clinical samples. *PAX* genes are a family of genes coding for transcription factors controlling developmental processes, with PAX5 implicated in neural development and B-cell differentiation [79,80]. In squamous cell carcinoma of the esophagus, the *PAX5* gene showed higher levels of promoter methylation and significantly downregulated mRNA expression compared with that of paired adjacent normal tissue [81]. Furthermore, *PAX5* gene methylation was associated with poor clinical outcomes, suggesting that *PAX5* methylation contributes to cisplatin resistance. In models of esophageal cancer cell lines, *PAX5* knockdown resulted in cisplatin resistance, supporting the notion of *PAX5* as a contributor to cisplatin resistance. In addition, knockdown studies revealed that *PAX5* expression was inversely correlated with the expression of *SLC2A1* [81], which encodes glucose transporter 1 and is also implicated in the cisplatin response [82], suggesting a possible mode of action. Altogether, the data suggest that *PAX5* gene methylation might predict cisplatin resistance and poor survival outcomes and may serve as a valuable diagnostic tool for cancer therapy of esophageal cancers. Correspondingly, in esophageal squamous cell carcinoma, the search for potential predictive factors influencing the response to chemotherapy identified the methylation of promoter CpG island of the *ZNF695* gene as one promising marker. The observed *ZNF695* methylation is associated with the response to cisplatin-based chemotherapy, suggesting that the methylation status of this gene might be a useful biomarker to predict the cisplatin response [83].

Epigenetic silencing of the putative DNA/RNA helicase Schlafen-11(SLFN11) has also been identified as a possible contributor to cisplatin resistance both in vitro and in clinical samples [84]. A comprehensive DNA methylation assay applied to the National Cancer Institute panel of 60 cancer cell lines revealed an association between *SLFN11* CpG promoter hypermethylation and increased resistance toward cisplatin and carboplatin. Furthermore, patients with ovarian and lung cancer had a less favorable clinical response to cisplatin or carboplatin chemotherapy when *SLFN11* CpG promoter hypermethylation was identified in the tumor tissues [84]. Regarding bladder cancer, for which cisplatin-based chemotherapy is among the most common regimens for treatment, it was shown that methylation of the promoter of p73, a homolog of p53, was associated with worse overall survival for patients receiving cisplatin-based chemotherapy, and it was therefore suggested that p73 promoter hypermethylation might be used as a prognostic indicator for the cisplatin response in bladder cancer patients [85].

Cisplatin-based chemotherapy is particularly successful in the treatment of testicular germ cell tumors (TGCTs) [1,86,87]. The biological basis of this exquisite cisplatin sensitivity is still unclear. Regarding clinical samples, early investigations revealed frequent promoter hypermethylation of various genes including *BRCA1* and *MGMT* in non-seminomatous TGCT, which was associated with gene silencing [88], while seminomas seldom showed promoter hypermethylation [89] suggesting that the epigenetic status may play a role in TGCT development and etiology. Furthermore, seminomas are more sensitive to cisplatin than non-seminomas and are severely hypomethylated, while non-seminomatous teratomas show the highest levels of methylation and are more difficult to treat [89]. Therefore, the epigenetic status might also influence cisplatin sensitivity and clinical outcomes for TGCTs. This is supported by investigations of Koul and co-workers, who observed different promoter hypermethylation of individual genes in cisplatin-sensitive versus resistant germ cell tumors [90].

On the other hand, DNA methylation might have a protective effect on cisplatin-induced toxicity. In the kidney, cisplatin causes cell injury and death in renal proximal tubule cells, leading to acute kidney injury (AKI), the main adverse effect of cisplatin treatment. Furthermore, 5-aza, which inhibits DNA methylation by blocking the transfer of the methyl group during DNA replication, increased cisplatin-induced apoptosis in rat renal proximal tubular cells, indicating a protective effect of methylation on cisplatin toxicity in renal cells [91]. Furthermore, genome-wide DNA methylation analysis revealed various differently methylated genes in the kidney tissue of mice treated with cisplatin compared with a control group, including genes involved in cell cycle control and apoptosis such as interferon regulatory factor 8 (Irf8). In addition, enhanced AKI following cisplatin treatment was observed in mice with *DNMT1* knockout in the kidney proximal tubules compared with a control group, indicating a reno-protective role of DNA methylation in vivo [91].

### 2.2. Histone Modification

Post-translational histone modification is another common pathway in epigenetics. These chromatin modifications can regulate the transcriptional machinery, in particular, through direct or indirect changes in the chromatin structure.

#### 2.2.1. Mechanism of Histone Modification

Histone proteins are structural proteins that form the core histone octamer consisting of two heterodimers of H2A and H2B together with two heterodimers of H3 and H4. In addition, 147 bp of DNA is wrapped around the core histones composing the structural unit of the nucleosome, while the histone protein H1 is responsible for the higher-order chromatin structure [92,93]. Histones are targets for various post-translational modifications, including methylation and acetylation (Figure 5). The amino- and carboxy-terminal tails of the histone proteins, which are rich in lysine and stick out of the nucleosome, are frequent sites of histone modification. The transcriptional status is linked to the addition or removal of different modifications such as methyl or acetyl groups to mainly lysine for acetylations and methylations also occurring on arginine residues. Alterations in the methylation and acetylation status influence the chromatin structure and allow or prevent access of transcription factors to DNA sequences, hence affecting gene expression [92,93]. Methylation of histones is controlled by histone methyltransferases (HMTs) and demethylases [94]. HMTs transfer methyl groups from SAM to lysine or arginine, with lysine methylation being relatively stable. Two families of HMTs have been identified, with the family of histone lysine methyltransferases (KMTs) including the enzymes enhancer of zeste homolog 2 (EZH2), G9a, DOT1L, and SETD2 [85], and the family of arginine methyltransferases (PRMTs) comprising PRMT1-9 [95]. HMTs might add up to three methyl groups to the amino acid, resulting in mono-, di-, and tri-methylated lysine or mono- and di-methylated arginine residues [94]. In this respect, lysine methylation is implicated in either transcriptional activation or gene repression, depending on which residue is modified and the type of modification. While tri-methylation of H3 lysine 4 (H3K4me3), H3 lysine 36 (H3K36me3), and H3 lysine 79 (H3K79me3) is linked to transcriptional activity, tri-methylation of H3 lysine 9 (H3K9me3) and H3 lysine 27 (H3K27me3) is associated with gene repression [96]. Methylation of histone arginine may also affect transcriptional activity, in both an activating and repressive manner [95]. Methylation of histone arginine residues appears to regulate developmental processes [97]. Furthermore, H3 arginine methylation (H3R17) has been implicated in inflammatory processes [97].

Along with histone methylation, acetylation of histone lysine residues is also recognized as an important epigenetic mechanism regulating gene expression. Histone acetylation is coordinated by two types of enzymes with reverse functions: histone acetyltransferases (HATs) and deacetylases (HDACs) [98,99]. HATs catalyze the transfer of a negatively charged acetyl group from cofactor acetyl-CoA to lysine residues in the N-terminal tail or the lateral surface of core histone proteins [100]. Lysine residues are positively charged, allowing interaction with the negatively charged DNA and resulting in a condensed chromatin structure. Acetylation of histone lysines leads to decondensation of the higher-order chromatin, which reflects the neutralization of lysine and hence decreased affinity to DNA, allowing access of transcription factors to DNA sequences and enabling transcriptional activation. Therefore, lysine acetylation predominantly correlates with chromatin accessibility and transcriptional activity. Hyper-acetylation of lysine tails is observed in transcriptionally active regions of chromatin, associated with transcriptional activity due to increased accessibility of transcription factors to the respective DNA sequence. HDACs, on the other hand, remove the acetyl groups from lysine, resulting in chromatin condensation and transcriptionally inactive chromatin, acting as transcriptional repressors [101,102]. Therefore, HATs and HDACs influence gene expression by an equilibrium between acetylation and de-acetylation.

#### 2.2.2. Histone Modification and Cisplatin Resistance

Members of the HMTs are involved in tumor development and progression and might also affect the efficacy of chemotherapeutics in cancer treatment [33]. A general overview of the influence of histone methlytransferases on drug resistance in cancer cells is given in [103]. Regarding cisplatin, the role of histone methyltransferases for chemotherapy resistance has been studied extensively both in vitro and in vivo models. It was observed that EZH2, a histone 3 lysine 27 (H3K27) methyltransferase, was overexpressed in cisplatin-resistant ovarian cancer cells compared with cisplatin-sensitive cells [104]. Knockdown of EZH2 decreased the level of H3K27 trimethylation (H3K27me3) and enhanced the sensitivity of cisplatin-resistant ovarian cancer cells and tumor xenografts to cisplatin, suggesting a role of EZH2 for cisplatin resistance probably through H3K27 methylation. This finding is in line with observations by Liu and co-workers, who found high expression of the methyltransferase G9a in cisplatin-resistant head and neck cancer cells (HNSCCs). Furthermore, high G9a expression was significantly associated with a poor chemotherapeutic response in HNSCC patients [105]. Increased cisplatin resistance has also been observed in a drug-tolerant subpopulation of the lung cancer cell line PC9, and this tolerance was accompanied by reduced levels of histone methylation (H3K4me3/2) together with reduced histone acetylation (H3K14ac), indicating that these epigenetic alterations affect cisplatin efficacy in lung cancer [106].

Regarding HATS and HDACs, aberrant expression has been discussed to result in epigenetic changes also leading to tumorigenesis and progression [107]. For instance, tissues of various cancers exhibit overexpression of HDACs, possibly affecting the expression of tumor suppressor genes and hence supporting malignant progress [108]. Furthermore, histone acetylation-dependent gene expression may also affect the chemosensitivity of cancers. A correlation between the overexpression of various types of HDACs and cisplatin resistance was shown in epithelial ovarian cancer cell lines [109]. Increased cisplatin resistance was also identified in lung cancer cell lines overexpressing actin-like 6A (ACTL6A), which is a component of several complexes implicated in chromatin remodeling, including the NuA4/TIP60 histone acetyltransferase [110]. The *ACTL6A* gene is frequently amplified in cancers, such as lung and ovarian cancer, and *ACTL6A* overexpression has been linked to an earlier relapse rate in ovarian and lung cancer patients who received cisplatin therapy. On a pre-clinical level, an association between the overexpression of *ACTL6A* and cisplatin resistance has been observed in lung and ovarian cancer cell lines and lung cancer xenografts, which has been explained by increased repair of cisplatin DNA adducts mediated by ACTL6A. However, unfortunately, no information about the acetylation status in the resistant cell lines and xenografts was presented in the study [110]. In another study, it was observed that HDAC1 was highly expressed in cisplatin-resistant ovarian cancer cells compared with a sensitive subline, also indicating reduced histone acetylation associated with cisplatin in ovarian cancer [111]. This was confirmed at the gene-specific level in a study by Cacan and co-workers, who showed that the enrichment of acetylated histone H3 was significantly lower at the *FAS* promoter in cisplatin-resistant A2780-AD ovarian cancer cells compared with chemo-sensitive A2780 cells, suggesting that the loss of *FAS* expression due to low acetylation contributes to cisplatin resistance [112]. This suggestion is supported by earlier findings revealing an association between the loss of *FAS* expression and resistance to chemotherapeutic drugs in ovarian cancer cells [113]. As clinical data showed low levels of H3/H4 acetylation accompanied by high HDAC activity in tissues of gastric cancer, a possible correlation between histone acetylation, HDAC activity, and the response to chemotherapy has been investigated in gastric cancer cell lines. These investigations revealed an inverse correlation between low levels of H3/H4 acetylation and high HDAC activity and the response to cisplatin [114].

### 2.3. Non-Coding RNAs

Along with DNA methylation and histone modification, which are the two major types of epigenetic gene silencing, RNA molecules are another class of epigenetic modulators. Several types of RNA occur, which can modify the chromatin such as miRNA, siRNA, and long non-coding RNA (lncRNA). In this respect, the interplay between these RNA molecules and RNA interference components is crucial for subsequent gene silencing. Additionally, the association between deregulated miRNA, siRNA, and levels is of relevance for cisplatin resistance. In the following, miRNA and lncRNA, two major modulators of gene expression, are discussed in detail.

#### 2.3.1. Mechanism of miRNA Induced RNA Interference

miRNAs are components of the RNA interference gene silencing pathway, which consists of various types of non-coding RNAs [115]. These small, highly conserved short-chain non-coding RNA molecules typically, ranging from 19–24 nucleotides in length, negatively regulate gene expression on the post-transcriptional level by binding to the 3′-untranslated region (3′-UTR) of their respective complementary target mRNAs. This binding may lead to degradation of the mRNA or inhibit its translation [115,116,117]. It should be noted that one miRNA is capable of targeting and controlling the expression of more than one mRNA and hence, more than one gene. miRNAs are important with respect to regulatory cellular processes, and there is increasing evidence that miRNAs are frequently dysregulated in most types of cancer influencing cell proliferation, invasion, metastasis, and epithelial–mesenchymal transition [118,119]. Furthermore, recent studies have connected the aberrant expression of miRNAs with chemoresistance in various cancer types [120]. In this respect, both increased levels of miRNA, as has been observed, for example, for miR-103, miR-645, or miR-223, or reduced expression, with examples including miR-7, miR-218, or miR-381, have been associated with chemoresistance.

#### 2.3.2. MicroRNAs and Cisplatin Resistance

The role of miRNA expression for cisplatin resistance has been studied extensively in both cancer cell lines and cancer tissues. Reduced levels of miR-218 have been reported in breast cancer-derived tumor tissues with increased cisplatin resistance as well as in a cisplatin-resistant MCF7 breast cancer subline, which was explained by the regulation of the tumor suppressor *BRCA1* by miR-218 [121]. Similarly, reduced levels of miRNA have been correlated with cisplatin resistance in various other tumor types. For lung cancer, Liu and co-workers reported that the level of miR-126-5p was reduced in cisplatin-resistant lung cancer cell lines and NSCLC tumor tissue [122]. In addition to downregulation, the upregulation of different miRNA levels is also observed in lung cancer. For instance, miR-7 sensitized cancer cells to cisplatin by enhancing apoptosis, and it was suggested that the miR-7 mediated decrease in the anti-apoptotic *BCL-2* gene contributed to increased cell death [123]. Furthermore, a better prognosis has been reported for patients with high levels of miR-7 expression in lung cancer tissues compared with patients with low expression levels of this type of miRNA. Unfortunately, no specific information with regard to the cisplatin response has been presented for the patients [123]. For laryngeal squamous cell carcinoma cells, on the other hand, miR-936 suppressed cell proliferation and increased cisplatin sensitivity, most likely by binding to its downstream target GPR78, which is connected to cell migration and metastasis [124]. Another recent analysis observed an association between miR-214-3p downregulation and chemo-resistance in cell lines and tissues of retinoblastoma, which is a common cancer in children [125]. Subsequent studies identified *XIAP* and *ABCB1* as targets of miR-214-3p, and it was suggested that 214-3p sensitizes retinoblastoma cells to chemodrugs and promotes apoptosis by targeting *XIAP* (X-linked inhibitor of apoptosis) and *ABCB1*, which codes for the multi-drug resistance P-glycoprotein.

Other studies reported an association between increased levels of miRNA and cisplatin resistance. A study by Luo and co-workers observed an association between enhanced expression of miR-103 and increased cisplatin survival due to reduced apoptosis in liver cancer cell lines [126]. As a target gene, *NOR1* was identified. Increased levels of miR-103 together with reduced levels of *NOR1* have also been observed in cisplatin-resistant hepatocellular carcinoma, indicating a clinical relevance of this type of miRNA [126]. Regarding colon cancer, increased levels of miR-645 have been associated with cisplatin resistance in both cancer cell lines and tumor tissues [127]. This can be explained by the finding that the expression of *SOX30*, which has been identified as a tumor suppressor gene [128], was negatively correlated with the expression of miR-645. Similarly, a possible link between miRNAs and chemoresistance has been reported in cancer-associated fibroblasts (CAFs) isolated from head and neck cancer tissue [129]. High levels of miR-196a were detected in cisplatin-resistant CAFs, together with low levels of CDKN1B and ING5, which are both implicated in the suppression of tumor growth. As target mRNAs of miR-196a, negative regulation of these genes might contribute to cisplatin resistance. For ovarian cancer cell lines, an association between miR-141 expression and cisplatin resistance has been observed, which was explained by the miR-141-mediated downregulation of *KEAP1* via mRNA degradation and translational repression [130]. However, no significant association between miR-141 levels and the response to cisplatin therapy was found in ovarian cancer patients. Nevertheless, the expression levels of the various miRNAs can be used as biomarkers for the cisplatin chemotherapy response. A thorough overview of miRNAs involved in cisplatin resistance in tumor tissue is presented by Loren and co-workers [120].

#### 2.3.3. Mechanism of lncRNA-Induced Gene Expression Alteration

Alongside miRNAs, lncRNAs are crucial factors in conferring cisplatin resistance across different tumor cell types. LncRNAs are commonly characterized as transcripts exceeding 200 nucleotides in length, with no protein-coding capacity, and are usually transcribed by RNA polymerase II, whereby few lncRNAs encode small proteins or micropeptides. Similar to mRNA, 5′-capping, splicing, and polyadenylation are similarly prevalent [131,132]. The expression of lncRNAs is cell type-specific, with higher cellular abundance compared with mRNA [133]. As for miRNAs, lncRNAs can be involved in epigenetic silencing or the activation of gene expression through intricate interactions with various cellular components. In contrast to miRNA, lncRNA exhibits a broader spectrum of reactivity, indirectly or directly interacting not only with mRNA but also with proteins, peptides, DNA, small-weight compounds, and even miRNA itself [131]. LncRNAs exhibit distinct and unique gene expression patterns in various cancer cells [134] and therefore represent a promising area of research in cancer therapy.

#### 2.3.4. Long Non-Coding RNAs and Cisplatin Resistance

LncRNAs play pivotal roles in numerous oncogenic pathways, and their dysregulation has the ability to affect cell survival. This section specifically addresses the involvement of lncRNAs in epigenetic-regulated chemoresistance. A comprehensive review also covering non-epigenetic cellular functions involved in chemoresistance is presented by Jiang and co-workers [135]. Recent in vivo and in vitro studies show that similar to miRNA, either the down- or upregulation of lncRNAs can affect key downstream targets, such as epigenetic regulators, leading to the modulation of gene expression [134].

One well-known lncRNA involved in cisplatin-induced chemoresistance is homeobox transcript antisense intergenic RNA (HOTAIR.) The overexpression of HOTAIR was first described in 2010 in breast cancer cells [136]. Since then, an enormous number of studies have been published that demonstrate abnormal expression of HOTAIR in various tumor types, among them, endometrial, lung, osteosarcoma, ovarian, and oral cancer [137]. An important function of HOTAIR is its ability to interact with essential epigenetic regulators such as polycomb repressive complex 2 (PRC2) and lysine-specific histone demethylase 1 A (LSD1) involved in gene silencing. HOTAIR acts as a scaffold for PRC2 and LSD1, which ultimately leads to target gene silencing through the formation of the gene inactivating histone mark H3K27me3 and a decrease in the activating mark H3K4me3 [137]. Furthermore, different studies show that HOTAIR is involved in cisplatin resistance mechanisms through its involvement in apoptotic pathways and cell cycle regulation. For example, the knockdown of HOTAIR in cisplatin-resistant osteosarcoma cells and tissue led to reduced expression of drug resistance-related genes and decreased resistance to cisplatin, suppressed cell proliferation and invasion, and enhanced apoptosis. Subsequent functional studies revealed that HOTAIR promoted cell apoptosis via miR-106a-5p/STAT3 signaling [138]. In addition, reducing HOTAIR expression with knockdown in cisplatin-resistant ovarian cancer cell lines resulted in induced chemosensitivity via the upregulation of miR-138-5p, a cancer suppressor, which is only expressed in the ovaries [139]. In addition, studies indicate that HOTAIR can not only interact directly with various miRNAs but can also influence other epigenetic regulatory mechanisms such as DNA methylation. In lung cancer cells, HOTAIR has been shown to reduce the methylation of homeobox A1 (HOXA1), a protein involved in cell proliferation and apoptosis, which is often associated with poor clinical outcomes when overexpressed. The reduced HOXA1 methylation status was linked to HOTAIR-mediated downregulation of DNMT1 and DNMT3b expression. Accordingly, the knockdown of HOTAIR increased chemosensitivity [140].

In addition, another lncRNA that is able to interact with different miRNAs contributing to chemoresistance is the HOXA transcript of the distal tip (HOTTIP). In gastric cancer patients, overexpression of HOTTIP is associated with a poor response to cisplatin therapy. In vitro studies show that a reduction in HOTTIP increases cisplatin sensitivity, for example, through the regulation of miR-218/HMGA signaling [141] and miR-216a-5p/BCL2/Beclin1/autophagy signaling [142]. Another known miRNA interactor is urothelial cancer-associated 1 (UCA1). The overexpression of UCA1 promotes tumor cisplatin resistance in some cancers, similar to the other lncRNAs discussed in this section. For example, higher levels of UCA1 were detected for oral squamous cell carcinoma and cisplatin-resistant cells. Functional assays revealed a repressive interaction with miR-184, which can regulate the expression of splicing factor 1. A subsequent in vivo study revealed that an increased cisplatin sensitivity of oral cancer cells could be caused by the knockdown of UCA1 [143]. Furthermore, the knockdown of UCA1 in gastric cancer cells enhanced cisplatin-induced apoptosis, possibly through its reduced ability to recruit EZH2 and to activate the PI3K/AKT signaling pathway, which is involved in tumor progression [144]. Apart from UCA1, taurine-upregulated gene 1 (TUG1) is another lncRNA that is involved in chemotherapy resistance. In bladder cancer cells, TUG1 possesses direct miRNA interaction capability and can regulate the EZH2 expression level. Investigations in bladder cancer tissues revealed that miR-194-5p is frequently downregulated due to promoter hypermethylation, and a cisplatin treatment can epigenetically increase this hypermethylation mediated by EZH2. Accordingly, the knockdown of TUG1 in bladder cancer cells lowers EZH2 expression, thereby reducing miR-194-5p methylation and inducing its expression. Accordingly, the downregulation of TUG1 leads to higher cisplatin sensitivity of bladder cancer cells [145]. Certainly, considering the numerous potential implications of lncRNA in carcinogenesis and chemoresistance, lncRNAs serve as a promising target for future cancer therapy.

## 3. Strategies to Restore Cisplatin Sensitivity

Due to the observation that cisplatin resistance is often associated with epigenetic modifications, one can envisage that epigenetic modifiers may potentially enhance the cytotoxicity of cisplatin and hence overcome the resistant phenotype. So far, two mechanisms of epigenetic therapy have been extensively studied in vitro and in clinical trials. Inhibitors of DNA methylation (DNMTi) and inhibitors of HDACs (HDACi) have shown a sensitizing effect on cisplatin toxicity in cell lines and anti-tumor activity in animal models or clinical samples.

### 3.1. Removal of DNA Hypermethylation by Targeting DNMTs

As cisplatin resistance is often associated with promoter hypermethylation of various genes, it has been proposed that reversal of promoter hypermethylation might restore cisplatin sensitivity. Specifically targeting hypermethylation can be achieved with the use of drugs with an inhibitory effect on DNMTs, such as 5-aza, decitabine, guadecitabine, and the newer drug zebularine (Zeb), which have shown promising anticancer effects. Treatment of ovarian cancer cells with the demethylating agent 5-aza sensitized the cells to cisplatin via the re-expression of the methylated *hSULF 1* gene, encoding a sulfate endosulfatase frequently downregulated in ovarian cancer [146]. In a preclinical study, it was shown that 5-aza–mediated reversal of promoter hypermethylation of several genes was accompanied by an increase in specific gene expression levels like *MLH1*, *GSTP1*, and *Casp8AP2* and resulted in the partial restoration of the cisplatin sensitivity in the platinum-resistant cervical cancer cell line SiHa S3 [147]. The sensitivity toward the platinating drug oxaliplatin was even restored with 5-aza to that observed in the parental platinum-sensitive SiHa cells. Similarly, 5-aza as an adjuvant agent to cisplatin reactivated *MHL1* expression and restored drug sensitivity in the cisplatin-resistant sublines of the A2780 ovarian cancer cell line [60]. In ovarian and colon tumor xenografts that are MLH1 negative because of promoter methylation, treatment with 5-aza led to a decrease in *MLH1* promoter methylation, which was associated with MLH1 expression and sensitized the xenografts to cisplatin [148]. This is in line with observations of Steele and co-workers, who also reported that 5-aza treatment resulted in a reversal of DNA methylation and re-expression of silenced genes, including *MLH1* and resensitized cisplatin-resistant cells and drug-resistant ovarian tumor xenografts [149]. As *p73* promoter methylation is associated with a poor cisplatin response in bladder cancer patients, several studies investigated whether inhibitors of methylation might affect cisplatin sensitivity in this tumor model. Indeed, treatment of bladder cancer cell lines with decitabine increased cisplatin sensitivity via a decrease in *p73* promoter methylation, which was associated with an increase in *p73* gene expression [85].

The DNMT inhibitor Zeb has also been successfully applied in cancer cell lines. Zeb-treated squamous cell carcinoma cells showed a strong reduction in methylated CpG islands together with an increase in cisplatin-induced toxicity [150]. Similarly, Zeb resensitized cisplatin-resistant ovarian cancer cells to the drug, most likely via the re-expression of the tumor suppressor gene *RASSF1A* [151].

Bladder cancer cell lines and bladder cancer tissue showed cisplatin resistance associated with CpG promoter hypermethylation of the *transcription factor homeobox (HOX) A9* gene [152]. Decitabine treatment reversed cisplatin resistance in bladder cancer cells, and it was therefore suggested that *HOXA9* could predict response to cisplatin-based chemotherapy in patients with bladder cancer. In addition to bladder cancer, decitabine is also known to improve cisplatin sensitivity in gastric cancer. One study attributes this to a reduction in DNA methylation at *sex-determining region Y-box 2 (SOX2*). Tissue of gastric cancer showed high expression levels of *SOX2*, and SOX2 overexpression is linked to a poor prognosis. Here, decitabine restored the expression of SOX2 by inducing *SOX2* DNA demethylation and a combined treatment with cisplatin synergistically inhibited the proliferation of xenograft tumors [153]. Another advanced demethylating agent is guadecitabine, with decitabine being its active metabolite [154]. Guadecitabine was already tested in combination with cisplatin in different clinical trials. For patients with platinum-refractory germ cell tumors, a five-day treatment with guadecitabine followed by cisplatin treatment demonstrated antitumor activity with a clinical benefit rate of 46% [155].

### 3.2. Manipulation of Histone Modifications as a Therapeutic Strategy in Cancer Treatment

HDACs are involved in different stages of tumor development and hence regulate tumorigenesis, and they also influence the efficacy of cisplatin treatment. For example, in cisplatin-resistant ovarian cancer cells, HDAC1 was highly expressed compared with cisplatin-sensitive cells and ovarian cancer xenografts [111]. Inhibition of HDACs has therefore been envisaged as a new tool in cancer therapy, and a combination of inhibitors of HDAC (HDACi) with cytostatics has demonstrated anti-tumor effects in preclinical cell culture models and clinical samples [156]. Various HDACi have been developed in recent years and intensively investigated alone or in combination with chemotherapeutic drugs and were found to significantly enhance the chemosensitivity of numerous cell lines and tumor types. HDACi are a group of small-molecule inhibitors exhibiting distinct specificity and activity, which can be grouped based on their chemical structure into (i) short-chain fatty acids such as sodium butyrate and valproic acid (VPA), (ii) organic hydroxamic acids including trichostatin A (TSA) and suberoylanilide hydroxamic acid (SAHA, vorinostat), and (iii) cyclic tetrapeptides such as trapoxin [157]. A summary of HDACi in cancer therapy is given by Suraweera and co-workers, who describe the effects of HDACi in combination therapy with various chemotherapeutic drugs in preclinical and clinical studies [158].

HDACi reverse deacetylation, which can increase the transcription and expression of silenced genes which might result in growth arrest and cell death. Sodium butyrate resulted in H3 acetylation and the expression of p53, which increased cisplatin sensitivity in two pancreatic cancer cell lines [159]. In human melanoma cells, VPA sensitized cells to cisplatin, for example, via increased expression of the cell cycle inhibitor p16 [160]. In ovarian cancer cell lines, VPA enhanced cisplatin-mediated toxicity [161], possibly via upregulation of the PTEN tumor suppressor phosphatase by VPA [162]. TSA sensitized drug tolerant sublines of the lung cancer cell line PC9 to cisplatin [106]. Similarly, vorinostat resensitized chemoresistant lymphoma cells to cisplatin and increased the toxicity of cisplatin in breast cancer cells [163]. As vorinostat increases histone acetylation by inhibiting HDACs, it was hypothesized that the more open chromatin structure due to histone acetylation results in better accessibility of the drug to DNA and hence, increased toxicity. Following VPA, TSA, and vorinostat treatment, increased binding of cisplatin to DNA has been observed in gastric cancer cell lines, together with cisplatin-induced cell death, possibly via increased expression of *p16*, *p21*, and *p27* [114]. A synergistic anticancer effect of VPA on cisplatin was also observed in gastric cancer xenograft models [114]. In a preclinical study, it was shown that the HDACi panobinostat can reverse cisplatin resistance, which was caused by overexpression of *ACTL6A* in lung cancer cell lines and xenografts [110]. The combination of cisplatin with the HDACi OSU-HDAC42 delayed tumor growth of ovarian cancer xenografts and prolonged animal survival, suggesting that OSU-HDAC42 can resensitize platinum-resistant ovarian tumors to cisplatin in vivo [164]. A combination of HDAC inhibition together with the inhibition of DNMT and AT-101, a small-molecule inhibitor of anti-apoptotic Bcl2 family members, increased cisplatin toxicity synergistically in various ovarian cancer cells [165].

### 3.3. Next-Generation Epigenetic Modifiers: RNA-Guided Precision Medicine

Non-coding RNAs, such as miRNAs and lncRNAs, are involved in different stages of carcinogenesis and multidrug resistance processes. In fact, they can act as tumor suppressors or tumor promoters affecting numerous genes and signaling pathways [166]. Hence, intensive research is being conducted on miRNA- and lncRNA-based therapies across a diverse spectrum of cancer types. Currently, different novel non-coding RNA-based therapies are being investigated in vitro and in vivo as promising cancer treatment options, for example, for hematological cancers, such as non-Hodgkin’s lymphoma [166] or lung cancers, among them NSCLC [167,168]. To overcome therapeutic resistance the combination of miRNA- and/or lncRNA-based therapy and chemotherapeutic agents, such as cisplatin, is currently under investigation [166,167,168]. Furthermore, an increasing number of epigenetic modifiers are being identified, offering potential applications in combination with chemotherapy to prevent or address chemoresistance. As an example, super-enhancers function as potent epigenetic modifiers that can overcome chemoresistance. They constitute a substantial cluster of diverse enhancers, exerting a compelling promotional influence on gene transcription activity [169].

## 4. Conclusions

Cisplatin is among the most effective anticancer drugs used to treat a broad spectrum of cancers. The clinical success, however, is limited by tumor cell resistance, which might be intrinsic to the tumor or acquired during cycles of therapy with cisplatin. Several factors contribute to drug resistance, including epigenetic modifications. Epigenetics is defined as somatic changes that are not associated with changes in DNA sequence. Numerous studies have shown that epigenetic dysregulation can affect the efficacy of cisplatin treatment (Figure 6). Aberrant changes in DNA methylation have been implicated in cisplatin resistance in vitro and in vivo. Regarding histone modifications, both methylation and acetylation have been associated with resistance to cisplatin, and this has been observed in pre-clinical and clinical samples. Aberrant miRNA and lncRNA expression have been identified in cisplatin-resistant cancer cell lines and cancer tissues, indicating miRNA and lncRNA expression as potential determinants of cisplatin resistance. Many experimental results have proved that the manipulation of epigenetic modifications is a promising approach to overcome cisplatin resistance, both in pre-clinical samples and in cancer tissues. Furthermore, the identification of epigenetic alterations in patients might be used as predictors for cisplatin response and hence, allow us to choose a suitable treatment.

## Figures and Tables

**Figure 1 ijms-25-01130-f001:**
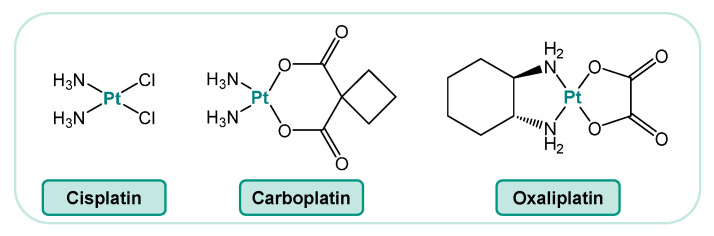
Chemical structures of the platinum-based anticancer drugs cisplatin, carboplatin, and oxaliplatin.

**Figure 2 ijms-25-01130-f002:**
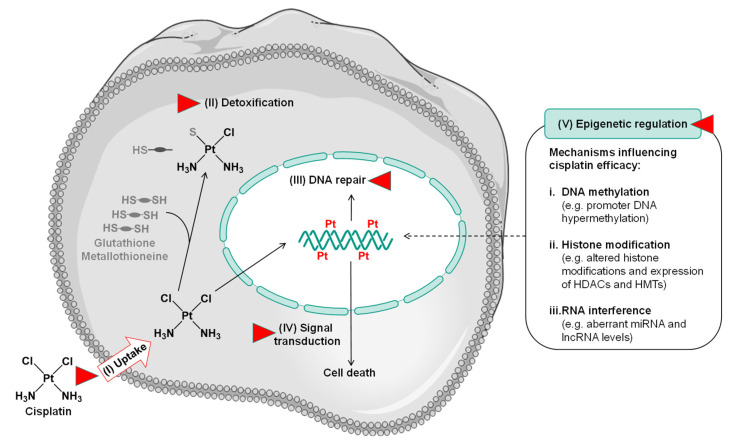
Mechanisms of resistance toward cisplatin. (I) Changes in efflux/uptake. (II) Increased detoxification due to enhanced levels of glutathione (GSH), glutathione-S-transferase (GST), or metallothioneins (MT). (III) Alterations in DNA repair. (IV) Alterations in the apoptosis pathway. (V) Changes in epigenetic regulation. HDAC: histone deacetyltransferase; HMT: histone methyltransferase; lncRNA: long non-coding RNA; miRNA: microRNA.

**Figure 3 ijms-25-01130-f003:**
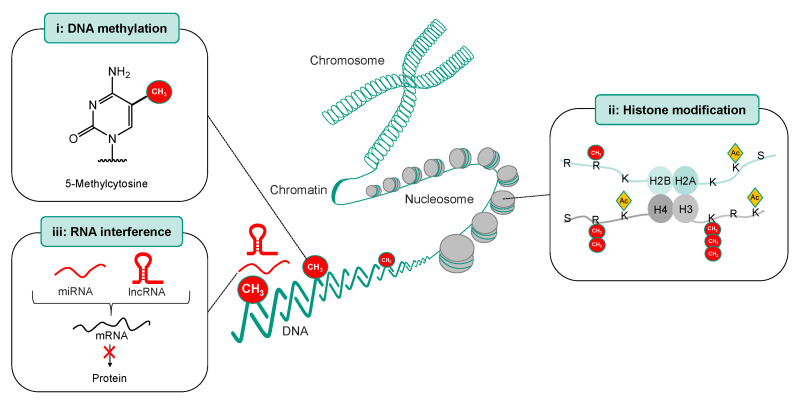
Mechanisms of epigenetic regulation. (i) DNA methylation. (ii) Histone modification. (iii) RNA interference, which can be mediated by non-coding RNA such as microRNA (miRNA) or long non-coding RNA (lncRNA).

**Figure 4 ijms-25-01130-f004:**
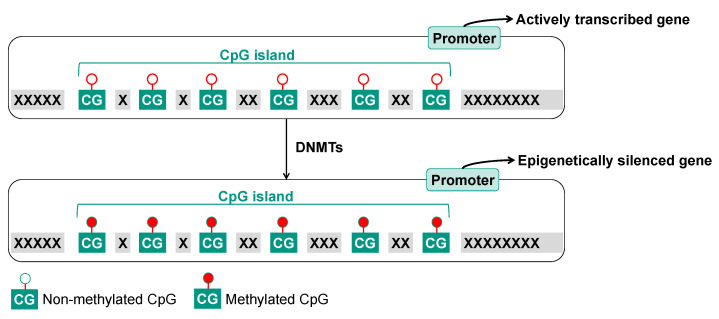
Regulation of gene expression by DNA methylation. Non-methylated CpG sites allow the transcription machinery access to the promoter region and hence, the expression of the respective gene. Methylated CpG sites silence the transcriptional machinery, resulting in the repression of gene transcription. DNMT: DNA methyltransferase.

**Figure 5 ijms-25-01130-f005:**
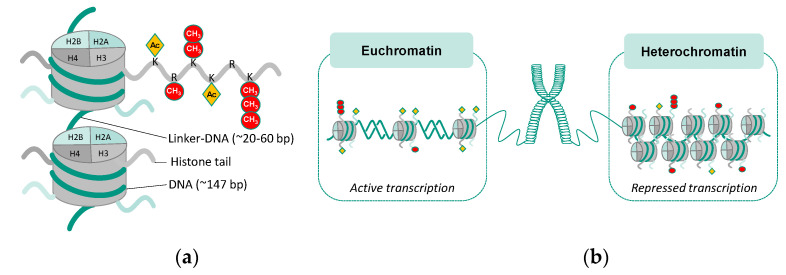
Post-translational histone modification and chromatin states. (**a**) Histone modification at lysine residues (K) and arginine residues (R) at the amino-terminal tail of the core histone proteins. (**b**) Histone modifications regulate chromatin structure and its corresponding transcriptional state.

**Figure 6 ijms-25-01130-f006:**
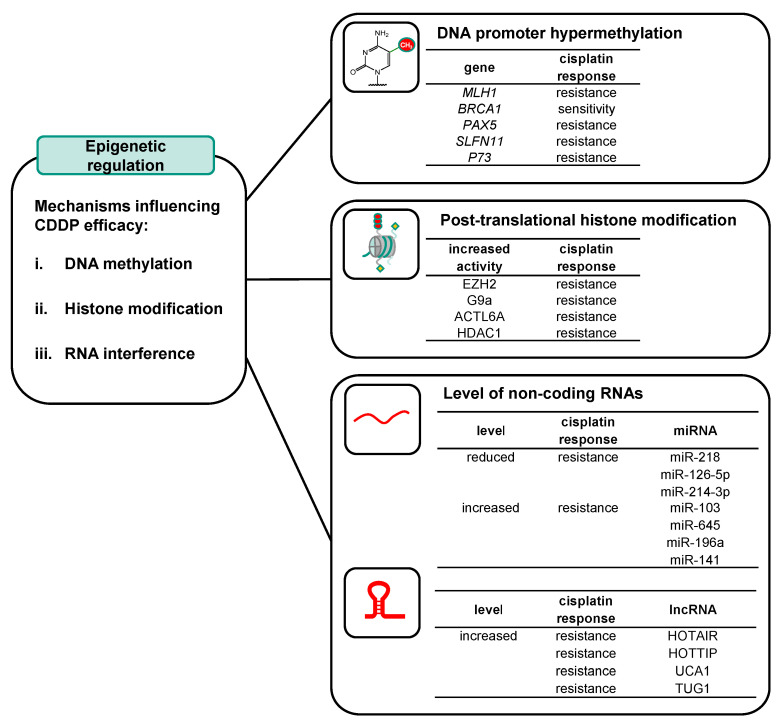
Selected mechanisms of epigenetic regulation affecting cisplatin efficacy. DNA promoter hypermethylation: MLH1 [59,60,61,62], BRCA1 [70,72,75], PAX5 [80,81], SLFN11 [84], p73 [85]; Post-translational histone modification: EZH2 [104], G9a [105], ACTL6A [110], HDAC1 [111]; miRNA level reduced: miR-218 [121], miR-126-5p [122], miR-214-3p [125], miRNA level increased: miR-103 [126]; miR-645 [127], miR-196a [129], miR-141 [130]; lncRNA level increased: HOTAIR [139,140], HOTTIP [141,142], UCA1 [143,144], TUG1 [145].

## Data Availability

Figure 2 was partly generated using Servier Medical Art, provided by Servier, licensed under a Creative Commons Attribution 3.0 unported license” (accessed on 16 October 2023). (https://creativecommons.org/licenses/by/3.0/).

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
