# Peer review of "Role of Epigenetics for the Efficacy of Cisplatin"

_ijms, 2024, doi:10.3390/ijms25021130_

Round 1

Reviewer 1 Report

Comments and Suggestions for Authors

Authors have presented the review very well. However few minor suggestions are provided.

Comments on the Quality of English Language

Language is fine. Only few typo errors.

Author Response

Reviewer 1:

We thank the reviewer for helpful comments and recommendations, which will improve the manuscript. Please find the following alterations in the maunscript:

Major comments:

  1. The abstract has been re-written with special emphasis given to epigenetics, it’s mechanisms, it’s usefulness as predictors of cisplatin response and epigenetic targeting therapies.
  2. The following references have been included: Einhorn LH. 2002 doi: 10.1073/pnas.072067999. de Vries G et al 2020 doi: 10.1016/j.ctrv.2020.102054. Cullen M et al doi: 10.1016/j.eururo.2019.11.022. PMID: 31901440; Pujol et al 2000 DOI 10.1054/bjoc.2000.1164
  3. Figure 6 contains a table on different epigenetic factors and their influence on cisplatin efficacy. Would the reviewer see the necessary to include a table in addition?
  4. An additional section (2.3.3. and 2.3.4.) has been included in the manuscript
  5. We exchanged some of the older papers with newer ones and added some new publications. We hope the reviewer is happy with our choice.

Minor comments:

  1. The manuscript has been adjusted as mentioned by the reviewer.
  2. The manuscript has been checked for typo errors.

Reviewer 2 Report

Comments and Suggestions for Authors

The manuscript “Role of epigenetics for the efficacy of cisplatin” is a review article overviewing the epigenetic modifications related to cisplatin efficacy. This review is excellent, is well written and is of interest for the readers of the journal; the aim of the study is clear and information are correctly presented. However, there are some major and minor concerns.

The following concerns should be addressed by authors in order to consider the manuscript suitable for publication.

Major

1.       Figure 1: authors use the abbreviation CDDP for cisplatin without mentioning it in the text. Please fix it. The same for Figure 5.

2.       Although the review is focused on cisplatin, I think would be useful to include in the introduction section also the other main platinum derivatives. Furthermore, a figure with their chemical structure is required.

3.       Line 25: cisplatin is very effective also in the treatment of breast cancer (PMID: 36897549).

4.       The introduction section describes only partially the mechanisms of action of cisplatin; beside the major mechanism explained in the review, cisplatin acts also as a ROS inducer, and massive ROS release results in the activation of apoptotic pathway (PMID: 36613780).

5.       The paragraph 2 does not provide a sufficient overview. One of the main concerns is that this manuscript ignores a master regulator of intracellular NAD and SAM content, the enzyme nicotinamide N-methyltransferase (NNMT), which can modulate the activity of enzymes involved in epigenetics, such as histone deacetylases sirtuins (PMID: 36829935). NNMT has been reported to be UPREGULATED in many tumors, where it contributes to the tumorigenicity and aggressiveness. Since NNMT can affect NAD homeostasis, NAD-dependent enzymes and concentration of SAM, it has a great impact on epigenetics, as demonstrated by Ulanovskaya et al. in an elegant study (PMID: 23455543). Notably, a number of NNMT inhibitors are already available and seems to be a promising strategy for overcoming chemoresistance (PMID: 34572571; PMID: 34704059; PMID: 34424711; PMID: 36104373). 

6.       Figure 5: it is not enough writing “reduced” or “increased” for the miRNAs. It must be specified in the figure which miRNA was analyzed in the study (e.g. miR-218 etc..)

7.       Most of studies reported are not properly discussed in their implications by authors.

8.       The authors should discuss the differences of mechanisms in cisplatin natural resistance and acquired resistance.

Minor

Line 427 and 581: “dys-regulated” should be written “dysregulated”.

Comments on the Quality of English Language

Minor editing of English language required

Author Response

Response to Reviewer 2

We highly appreciate that the reviewer found the manuscript well written and of interest to readers of the special issue and we would like to thank for the helpful comments to improve the quality of our paper.

Major concerns:

  1. The abbreviation CDDP in Figures 1 and 5 in now explained in the subtitle and main text.
  2. Carboplatin and oxaliplatin are now included in the introduction section (“Cisplatin and its analogues carboplatin and oxaliplatin are platinum-containing drugs that are widely used in cancer therapy. Cisplatin (cis-diamminedichloro-platinum(II)) is a major component for the treatment of a variety of malignancies and most effective particularly for patients with testicular germ cell tumors or ovarian cancers, but it also displays clinical activity against bladder, prostate, head and neck, cervical, breast and lung cancers. Carboplatin shows a comparable mode of action and is used as first-line treatment of patients with advanced ovarian cancer, and it is also applied for the treatment of a number of other types of cancer, such as advanced small cell and non-small cell lung cancer. Oxaliplatin, which has a distinct pattern of activity, is used in combination with 5-fluorouracil for the treatment of metastatic colorectal cancer, which is insensitive to treatment with cisplatin and carboplatin“). A figure with the chemical structures of the platinum drugs has been added (Figure 1).
  3. The information and reference have been included
  4. The information “Besides damaging DNA cisplatin is also known to induce reactive oxygen species which in turn will result in the activation of apoptotic pathways and hence contribute to cisplatin cytotoxicity“ has been included in the manuscript.
  5. In paragraph 2 NNMT has been included („The celluar methylation status is also regulated by expression and activity of the nicotinamide N-methyltransferase (NNMT). NNMT is an enzyme which catalyzes the methylation of nicotinamide to N-methyl nicotinamide by using S-adenosyl methionine (SAM) as methyl group donor. It has been observed that NNMT expression and activity are frequently elevated in tumor tissue causing a decrease in the cellular SAM content which in turn will lead to hypomethylation of genes associated with tumor progression and metastasis. As NNMT supports tumorigenesis it therefore may be considered a potential therapeutic anticancer target. Indeed various molecules that target NNMT have been identified, which might be applied to overcome chemoresistane“).
  6. Figure 5 has been changed according to the reviewer`s request.
  7. We are sorry that the reviewer feels that “most of the studies are not properly discussed in their implications”. We would ask the reviewer for an example as we do not understand the criticism. Does the reviewer want a thorough discussion of every study? We are afraid that would go beyond the scope of a review, but we are happy to make the changes asked for.
  8. The sentence has now be changed to: Intrinsic resistance is often associated with inactivation of the DNA mismatch repair process (MMR), which is observed in inherently cisplatin resistant colorectal cancer. Acquired cisplatin resistance has been extensively studied in experimental models of cisplatin resistant cell lines.

Minor concern:

  1. Dys-regulated has been changed to dysregulated.

Round 2

Reviewer 2 Report

Comments and Suggestions for Authors

The authors properly addressed all the concerns raised by the reviewer and therefore the manuscript deserves to be published.

Comments on the Quality of English Language

Minor editing of English language required